# AdvectiveNet: An Eulerian-Lagrangian Fluidic reservoir for Point Cloud Processing

**Xingzhe He**
Dartmouth College
Rutgers University [*]

**Helen Lu Cao**
Dartmouth College [†]

**Bo Zhu**
Dartmouth College [‡]

## Abstract

This paper presents a novel physics-inspired deep learning approach for point cloud processing motivated by the natural flow phenomena in fluid mechanics. Our learning architecture jointly defines data in an Eulerian world space, using a static background grid, and a Lagrangian material space, using moving particles. By introducing this Eulerian-Lagrangian representation, we are able to naturally evolve and accumulate particle features using flow velocities generated from a generalized, high-dimensional force field. We demonstrate the efficacy of this system by solving various point cloud classification and segmentation problems with state-of-the-art performance. The entire geometric reservoir and data flow mimics the pipeline of the classic PIC/FLIP scheme in modeling natural flow, bridging the disciplines of geometric machine learning and physical simulation.

## 1 Introduction

The fundamental mechanism of deep learning is to uncover complex feature structures from large data sets using a hierarchical model composed of simple layers. These data structures, such as a uniform grid (Lecun et al., 1998), an unstructured graph (Kipf & Welling, 2016), or a hierarchical point set (Qi et al., 2016a; 2017), function as geometric reservoirs to yield intricate underpinning patterns by evolving the massive input data in a high-dimensional parameter space. On another front, computational physics researchers have been mastering the art of inventing geometric data structures and simulation algorithms to model complex physical systems (Gibou et al., 2019). Lagrangian structures, which track the motion in a moving local frame such as a particle system (Monaghan, 1992), and Eulerian structures, which describe the evolution in a fixed world frame such as a Cartersian grid (Fedkiw et al., 2001), are the two mainstream approaches. Various differential operators have been devised on top of these data structures to model complex fluid or solid systems.

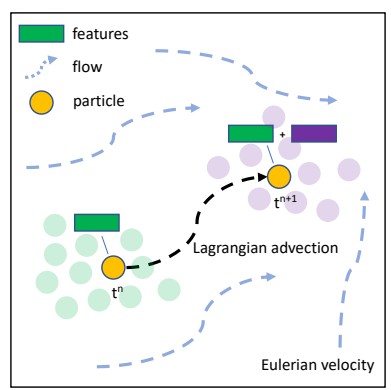

Figure 1: We build an advective network to create a fluidic reservoir with hybrid Eulerian-Lagrangian representations for point cloud processing.

Pioneered by E (2017) and popularized by many others, e.g., (Long et al., 2018; Chen et al., 2018; Ruthotto & Haber, 2018), treating the data flow as the evolution of a dynamic system is connecting machine learning and physics simulation. As E (2017) notes, there exists a mathematical equivalence between the forward data propagation on a neural network and the temporal evolution of a dynamic system. Accordingly, the training process of a neural network amounts to finding the optimal control forces exerted on a dynamic system to minimize a specific energy form.

Point cloud processing is of particular interest under this perspective. The two main challenges: to build effective convolution stencils and to evolve learned nonlinear features (Qi et al., 2016a;

---

[*]xingzhe.he@rutgers.edu

[†]Helen.L.Cao.22@dartmouth.edu

[‡]bo.zhu@dartmouth.edu

Atzmon et al., 2018; Wang et al., 2019), can map conceptually to the challenges of devising world-frame differential operators and tracking material-space continuum deformations when simulating a PDE-driven dynamic system in computational physics. We envision that the key to solving these challenges lies in the adaption of the most suited geometric data structures to synergistically handle the Eulerian and Lagrangian aspects of the problem. In particular, it is essential to devise data structures and computational paradigms that can accommodate global fast convolutions, and at the same time track the non-linear feature evolution.

The key motivation of this work originates from physical computing that tackles its various frame-dependent and temporally-evolved computational challenges by creating the most natural and effective geometric toolsets under the two different viewpoints. We are specifically interested in uncovering the intrinsic connections between a point cloud learning problem and a computational fluid dynamic (CFD) problem. We observe that the two problems share an important common thread regarding their computational model, which both evolve Lagrangian particles in an Eulerian space guided by the first principle of energy minimization. Such observations shed new insight into the 3D point cloud processing and further opens the door for marrying the state-of-the-art CFD techniques to tackle the challenges emerging in point cloud learning.

To this end, this paper conducts a preliminary exploration to establish an Eulerian-Lagrangian fluidic reservoir that accommodates the learning process of point clouds. The key idea of the proposed method is to solve the point cloud learning problem as a flow *advection* problem jointly defined in a *Eulerian* world space and a *Lagrangian* material space. The defining characteristic distinguishing our method from others is that the spatial interactions among the Lagrangian particles can evolve temporally via *advection* in a learned flow field, like their fluidic counterpart in a physical circumstance. This inherently takes advantage of the fundamental flow phenomena in evolving and separating Lagrangian features non-linearly (see Figure 1). In particular, we draw the idea of Lagrangian advection on an Eulerian reservoir from both the Particle-In-Cell (PIC) method (Evans & Harlow, 1957) and the Fluid-Implicit-Particle (FLIP) method (Brackbill et al., 1987), which are wholly recognized as 'PIC/FLIP' in modeling large-scale flow phenomena in both computational fluids, solids, and even visual effects. We demonstrate the result of this synergy by building a physics-inspired learning pipeline with straightforward implementation and matching the state-of-the-art with this framework.

The key contributions of our work include:

- An advective scheme to mimic the natural flow convection process for feature separation;
- A fluid-inspired learning paradigm with effective particle-grid transfer schemes;
- A fully Eulerian-Lagrangian approach to process point clouds, with the inherent advantages in creating Eulerian differential stencils and tracking Lagrangian evolution;
- A simple and efficient physical reservoir learning algorithm.

## 2 RELATED WORKS

This section briefly reviews the recent related work on point cloud processing. According to data structures used for building the convolution stencil, the methods can be categorized as Lagrangian (using particles only), Eulerian (using a background grid), and hybrid (using both). We also review the physical reservoir methods that embed network training into a physical simulation process.

**Lagrangian** Lagrangian methods build convolution operators on the basis of local points. Examples include PointNet (Qi et al., 2016a), which conducts max pooling to combat any disorganized points, PointNet++ (Qi et al., 2017), which leverages farthest point sampling to group particles, and a set of work (Wang et al., 2019; Xu et al., 2018; Li et al., 2018b;a; Jiang et al., 2018) based on k-nearest neighbors. Beyond the mesh-free approaches, researchers also seek to build effective point-based stencils by establishing local connectivities among points. Most significantly, geometric deep learning (Bruna et al., 2013; Bronstein et al., 2016) builds convolution operators on top of a mesh to uncover the intrinsic features of objects' geometry. In particular, we want to highlight the work on dynamic graph CNN (Wang et al., 2019), which builds directed graphs in an extemporaneous fashion in feature space to guide the point neighbor search process, which shares similarities with our approach.

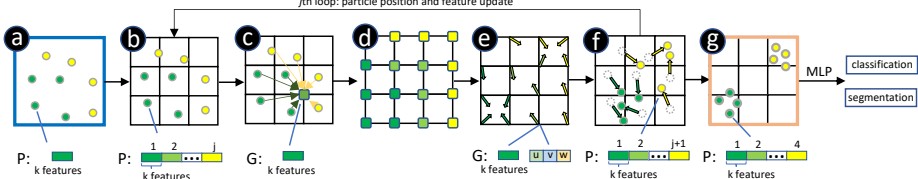

Figure 2: **Workflow overview:** a) The feature vector for each particle is initialized by a $1 \times 1$ convolution; b) Particles are embedded in an Eulerian grid; c) Features are interpolated from particles to the grid, denoted as $\boldsymbol{I}_P^G$; d) 3D convolution is applied on the grid to calculate the generalized forces and grid features; e) A velocity field is generated on the background grid; f) Particles advect in the Eulerian space using the interpolated velocities; grid features are interpolated to particles, denoted as $\boldsymbol{I}_G^P$, and appended to its feature vector; g) Particles aggregate. The workflow consists of one loop to update the particle positions and features iteratively with temporal evolution. Finally, the Lagrangian features are fed into a fully-connected network for classification and segmentation.

**Eulerian** Eulerian approaches leverage background discretizations to perform computation. The most successful Eulerian method is the CNN (Lecun et al., 1998), which builds the convolution operator on a 2D uniform grid. This Eulerian representation can be used to process 3D data by using multiple views (Su et al., 2015; Qi et al., 2016b; Feng et al., 2018) and extended to 3D volumetric grids (Maturana & Scherer, 2015; Qi et al., 2016b; Z. Wu, 2015). Grid resolution is the main performance bottleneck for 3D CNN methods. Adaptive data structures such as Octree (Riegler et al., 2016; Wang et al., 2017; 2018), Kd-tree (Klokov & Lempitsky, 2017), and multi-level 3D CNN (Ghadai et al., 2018) were invented to alleviate the problem. Another example of Eulerian structures is Spherical CNN (Cohen et al., 2018) that projects 3D shapes onto a spherical coordinate system to define equivalent rotation convolution. In addition to these voxel-based, diffusive representations, shapes can also be described as a sharp interface modeled as an implicit level set function (Hu et al., 2017; Park et al., 2019; Mescheder et al., 2019). For each point in the space, the level set function acts as a binary classifier checking whether the point is inside the shape or not.

**Hybrid** There have been recent attempts to transfer data between Lagrangian and Eulerian representations for efficient convolution implementation. These data transfer methods can be one-way (Wang et al., 2017; Klokov & Lempitsky, 2017; Tchapmi et al., 2017; Le & Duan, 2018), in which case the data is mapped from points to grid cells permanently, or two-way (Su et al., 2018; Atzmon et al., 2018; Liu et al., 2019; Groueix et al., 2018), in which case data is pushed forward from particle to grid for convolution and pushed backward from grid to particle for evolution. Auto-encoders on point clouds (Fan et al., 2016; Achlioptas et al., 2017; Yang et al., 2017; Yu et al., 2018; Zhao et al., 2019) can be also regarded as a hybrid approach, where encoded data is Eulerian and decoded data is Lagrangian. In addition, we want to mention the physical reservoir computing techniques that focus on the leverage of the temporal, physical evolution to solve learning problems, e.g., see (Jaeger, 2001) and (Maass et al., 2002). Physical reservoir computing is demonstrating successes in various applications (Jalalvand et al., 2015; Jaeger, 2002; Hauser et al., 2012; Lukoševičius & Jaeger, 2009; Tanaka et al., 2019).

## 3 ALGORITHM

**PIC/FLIP overview** Before describing the details of our method, we begin with briefly surveying the background of the PIC/FLIP method. PIC/FLIP uses a hybrid grid-particle representation to describe fluid evolution. The particles are used for tracking materials, and the grid is used for discretizing space. Properties such as mass, density, and velocity are carried on particles. Each simulation step consists of four substeps: particle-to-grid transfer $\boldsymbol{I}_P^G$, grid force calculation (*Projection*), grid-to-particle transfer $\boldsymbol{I}_G^P$, and moving particles (*Advection*). In the $\boldsymbol{I}_P^G$ step, the properties on each particle are interpolated onto a background grid. In the *Projection* step, calculations such as adding body forces and enforcing incompressibility are conducted on the background grid. After this, the velocities on grid nodes are interpolated back onto particles, i.e., $\boldsymbol{I}_G^P$. Finally, particles move to their new positions for the next time step using the updated velocities (*Advection*). As summarized above, the key philosophy of PIC/FLIP is to carry all features on particles and to perform all differential

calculations on the grid. The background grid functions as a computational paradigm that can be established extemporaneously when needed. Data will transfer from particle to grid and then back to particle to finish a simulation loop.

Our proposed approach follows the same design philosophy as PIC/FLIP by storing the learned features on particles and conducting differential calculations on the grid. The Lagrangian features will evolve with the particles moving in an Eulerian space and interact with local grid nodes. As shown in Figure 2, the learning pipeline mimics the PIC/FLIP simulation loop in the sense that Lagrangian particles are advected passively in an Eulerian space guided by a learned velocity field.

**Initialization** We initialize a particle system $\boldsymbol{P}$ and a background grid $\boldsymbol{G}$ as the Lagrangian and Eulerian representations respectively for processing point clouds. We use the subscript $p$ to refer to particle indices and $i$ to refer to the grid nodes. For the Lagrangian portion, the particle system has $n$ particles, with each particle $P_p$ carrying its position $\boldsymbol{x}_p \in \mathbb{R}^3$, velocity $\boldsymbol{v}_p \in \mathbb{R}^3$, mass $m_p \in \mathbb{R}$, and a feature vector $\boldsymbol{f}_p \in \mathbb{R}^k$ ($k = 64$ initially). The particle velocity is zero at the beginning. The particle mass $m_p = 1$ will keep constant over the entire evolution. To initialize the feature vector $\boldsymbol{f}_p$, we first put the particles in a grid with size $N^3$. For each cell, we calculate 1) the center of mass of all the particles in the cell, and 2) the normalized vector pointing from each particle to the this mass center. For each particle, we concatenate these two vectors to the initial feature vector. This process is repeated for $N = 2, 4, 6, 8, 10, 12$. The resulting feature vector with the length of $6 \times 6$ are fed into a multi-layer perceptron (MLP) to generate the feature vector $\boldsymbol{f}_p$.

For the Eulerian part, we start with a 3D uniform grid $\boldsymbol{G}$ to represent the bounding box of the particles. The resolution of the grid is $N^3$ ($N = 16$ for most of our cases). At the beginning, the particle system and its bounding box are normalized to the space of $[-1, 1]^3$. Each grid node $G_i$ of $\boldsymbol{G}$ stores data interpolated from the particles.

**Particle-grid transfer** Both the interpolation from grid to particle and particle to grid are executed using tri-linear interpolation, which is a common scheme for property transfer in simulation and learning code.

**Generalized grid forces** With the feature vectors transferred from particles to grid nodes, we devise a 3D CNN on the grid to calculate a generalized force field based on the Eulerian features. The network consists of three convolution layers, with each layer as a combination of 3D convolution, batch norm, and ReLU. The input of the network is a vector field $\mathbf{F}^{(kj) \times N \times N \times N}$ composed of the feature vectors on all grid nodes, with $k$ as the feature vector size (64 by default) and $j$ as the iteration index in the evolution loop (see Figure 2). The output is a convoluted vector field $\mathbf{F}_c^{(kj) \times N \times N \times N}$ with the same size as $\mathbf{F}$.

We use $\mathbf{F}_c$ for two purposes: 1) To interpolate $\mathbf{F}_c$ from the grid back onto particles and append it to the current feature vector in order to enrich its feature description; 2) To feed $\mathbf{F}_c$ into another single-layer network to generate the new Eulerian velocity field $\mathbf{V}$ for the particle advection. Specifically, this $\mathbf{V}$ is interpolated back onto particles in the same way as the feature interpolation to update the particle positions for the next iteration (see Advection for details).

**Advection** The essence of an advection process is to solve the advection equation with the Lagrangian form $D\boldsymbol{v}/Dt = 0$ or the Eulerian form $\partial \boldsymbol{v}/\partial t + \boldsymbol{v} \cdot \nabla \boldsymbol{v} = 0$. The advection equation describes the passive evolution of particle properties within a flow field. With the learned grid velocity field in hand, we will update the particle velocity following the conventional scheme of PIC/FLIP. Specifically, the new velocity is first updated by interpolating the Eulerian velocity to particles (the PIC step):

$$\boldsymbol{v}_{PIC}^{n+1} = \boldsymbol{I}_G^P(\boldsymbol{v}_g^{n+1}) \tag{1}$$

Then, we interpolate the difference between the new and the old Eulerian velocity:

$$\boldsymbol{v}_{FLIP}^{n+1} = \boldsymbol{v}_p^n + \boldsymbol{I}_G^P(\boldsymbol{v}_g^{n+1} - \boldsymbol{I}_P^G(\boldsymbol{v}_p^n)), \tag{2}$$

and then add them to the particle with a weight $\alpha$ (=0.5 in default.):

$$\boldsymbol{v}_p^{n+1} = \alpha * \boldsymbol{v}_{PIC}^{n+1} + (1 - \alpha) * \boldsymbol{v}_{FLIP}^{n+1} \tag{3}$$

With the updated velocity on each particle from the $\boldsymbol{I}_G^P$ interpolation, the particle's position for the next time step can be updated using a standard time integration scheme (explicit Euler in our implementation):

$$\boldsymbol{x}_p^{n+1} = \boldsymbol{x}_p^n + \boldsymbol{v}_p^{n+1}\Delta t. \tag{4}$$

**Boundary conditions**   We apply a soft boundary constraints by adding an penalty term in the objective function to avoid particles moving outside of the grid:

$$\phi_b = \frac{1}{n}\sum_p max(0, \|x_p\|_2 - 1) \tag{5}$$

where $x_p$ represents the $p$th particle in the whole batch and $n$ is the number of particles in the whole batch. We penalize on all the particles that run outside the grid.

We also design the gather penalty and the diffusion objectives to enhance the particle diffusion and clustering effects during evolution (specifically for the segmentation application):

$$\phi_g = \frac{1}{2}\sum_l\sum_m max(0, 1 - \|c_l - c_m\|) \tag{6}$$

$$\phi_d = \frac{1}{n}\sum_l\sum_p \|c_l - x_{lp}\| \tag{7}$$

where $c_l$ and $c_m$ are the centers of particles of label $l$ and $m$ and $x_{lp}$ is the $p$th particle with label $l$.

## 4   NETWORK ARCHITECTURE

The global architecture of our network is shown in Figure 3. Our model starts from a point cloud with the position of each point. After an initialization step ending with a two-layer MLP (64,64), each point carries a feature vector of length 64. These features are fed into the advection module to exchange information with neighbors. The generated features have two uses: to generate the velocity for each particle, and to be used along with the new advected particle position to collect information from neighbors. This process repeats for a few times to accumulate features in the feature space and to aggregate particles in the physical space.

**Advection module**   The data flow inside the advection module starts with particles, passes through layers of grids, then sinks back to particles. This module takes the position and the feature vector as input. The feature vectors are first fed into an MLP to reduce its dimensions to 32, which saves computational time and prevents over-fitting. Then, we apply three layers of convolution that are each a combination of 3D convolution, batch norm, and ReLU, with a hidden-layer size as (32,16,32) on the grid, to obtain a high-dimensional, generalized force field on the grid. Afterwards, a velocity field is generated from this force field by another two-layer network. The velocity field is then interpolated back to particles for Lagrangian advection. Additionally, to generate the output feature vector, the input and output features (with 32-dimension each) are concatenated together and appended to the original feature vector. The output of the advection module is a set of particles with new positions and new features that are ready to process for the next iteration as in Figure 2.

## 5   EXPERIMENTS

We conducted three parts of experiments, including the ablation tests and the applications for classification and segmentation. We implemented the system in PyTorch (see the submitted source code) and conducted all the tests on a single RTX 2080 Ti GPU. In the ablation tests, we evaluated the functions of the advection module, temporal resolution, grid resolution, and the functions of the PIC/FLIP scheme on ModelNet10 (Z. Wu, 2015) and ShapeNet (Yi et al., 2016). For classification, we tested our network on ModelNet40 and its subset ModelNet10. We used the class prediction accuracy as our metric. For segmentation, we tested our network on ShapeNet (Yi et al., 2016) and S3DIS data set (Armeni et al., 2016). We used mean Intersection over Union (mIoU) to evaluate our method and compare with other benchmarks.

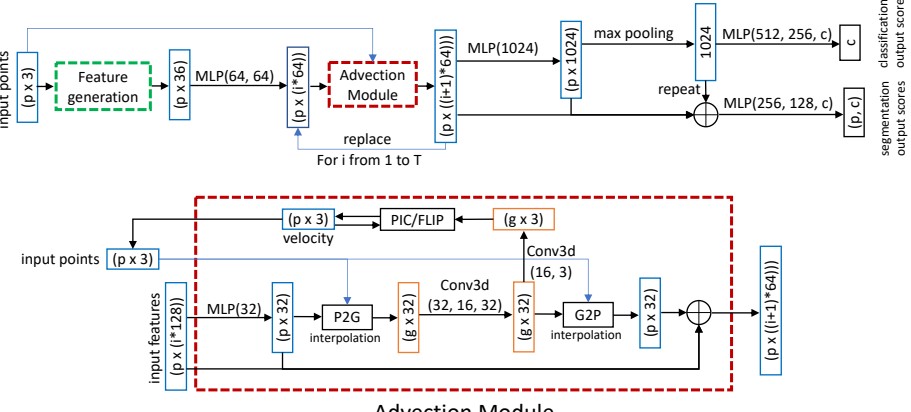

Advection Module

Figure 3: **Network architectures:** The *top* diagram demonstrates the global architecture of our network with detailed information for tensor dimensionality and modular connectivity. The blue box is for particle states and the orange box indicates grid states. The dotted green box is the module generating the initial Lagrangian features. The dotted red box is for the functional module of advection (see the bottom diagram). The states are connected with multi-layer perceptrons (black arrows in the diagram). Each MLP has a number of hidden layers with a different number of neurons (specified by the numbers within the parentheses). The *bottom* figure shows the details of the advection module updating the particle *features* by transferring data on the grid and concatenating particles. Meanwhile, it updates the particle *positions* with the generalized Eulerian forces calculated on the grid.

## 5.1 ABLATION EXPERIMENTS

**Advection**    We turn off the advection module to verify the its effectiveness for the final performance. We conducted the comparison on the ShapeNet data set (Yi et al., 2016). The mIoU reached $86.2\%$ with the advection module in comparison to $85.3\%$ without it, necessitating the role of the advection step.

**Temporal resolution**    *(Physical Intuition)* The evolution of a dynamic system can be discretized on the temporal axis by the numerical integration with a number of steps. Given a fixed total time, the number of timesteps is in an inverse ratio to the length of each step. For a typical explicit scheme (e.g., explicit Euler), a small timestep leads to a numerically secure result at the expense of performing more time integrations; while a large timestep, although efficient, might explode out of the stable region.

Table 1: Temporal resolution

| # ts | 0 | 1 | 2 | 3 | 4 | 5 | 6 | 7 | 8 |
|---|---|---|---|---|---|---|---|---|---|
| Acc | 93.2 | 95.2 | 95.4 | 94.8 | 94.7 | 95.1 | 95.1 | 95.2 | 95.1 |

*(Numerical Tests)* Motivated by this numerical intuition, we investigated the effects of temporal resolution on our learning problem. Specifically, we tested the performance of the network regarding both the learning accuracy and the evolved shape by subdividing the numerical integration into 0-8 steps (0 means no integration). The test was performed on ModelNet10. As shown in Table 1 and Figure 4, the learning accuracy stabilizes around

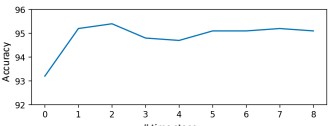

Figure 4: Temporal accuracy

$95\%$ as the number of integration increases, with 2 steps and 4 steps as the maximum ($95.4\%$) and minimum ($94.7\%$), indicating a minor effect from the temporal resolution on learning accuracy. For the shape convergence, we demonstrated that different temporal resolutions converge to very similar final equilibrium states, despite of the different time step sizes. As shown in Figure 5, the point-cloud model of an airplane is advected with different velocity fields generated on different temporal resolutions. The final shapes with timestep 2, 3, and 6 all exhibit the same geometric feature separations and topological relations. This result evidences our conjecture that all the temporal resolutions we used are within the stable region, motivating us to pick a larger time step size (total time/3 for most of our cases) for efficiency.

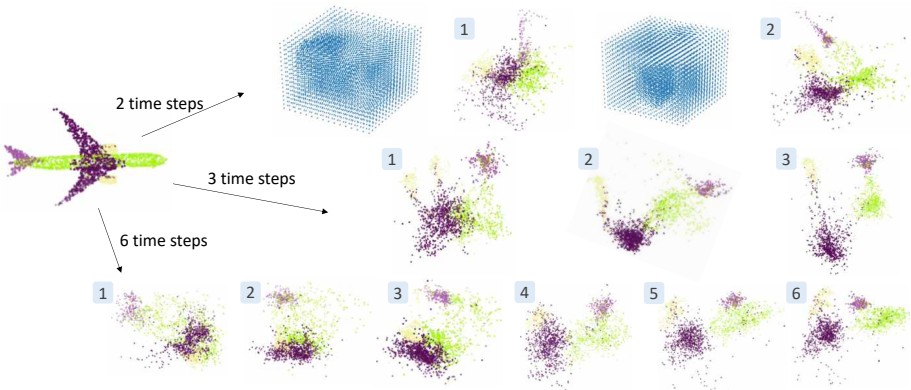

Figure 5: Visualization of the advection of an airplane is shown with time steps of 2, 3 and 6. Note that we rotate the point cloud and normalize the velocity field for visualization purposes.

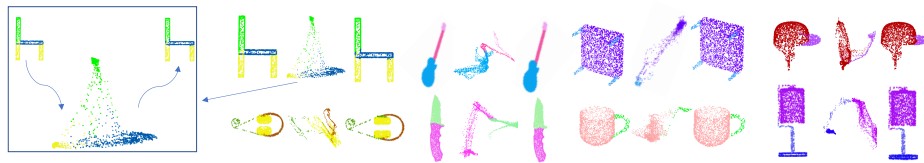

Figure 6: **Visualization of segmentation**. Examples of different categories are depicted, consisting of initial shape, intermediary grouping, and final part prediction.

Table 2: Spatial resolution

|  | $8^3$ | $16^3$ | $32^3$ |
|---|---|---|---|
| ModelNet10 Acc (1024 pnts) | 94.4 | **95.4** | 95.1 |
| ModelNet10 pnts per cell | 6.8 | **1.6** | 1.0 |
| ShapeNet mIoU (2048 pnts) | 85.4 | 86.1 | **86.2** |
| ShapeNet pnts per cell | 21.4 | 5.2 | **1.7** |

**Spatial resolution**   *(Physical Intuition)* For a typical particle-grid simulation in CFD, the resolution of the grid and the number of particles are correlated. Making sure that each grid cell should contain enough number of particles (e.g., 1-2 particles per cell), ensures information exchange between these two discretizations is accurate. Empirically, an overly refined grid will lead to inaccurate Eulerian convolution due to the large bulk of empty cells, while an overly coarse grid will dampen the motion of particles due to artificial viscosity (e.g., see Evans & Harlow (1957); Brackbill et al. (1987)), which makes the number of particles per cell $ppc$ a key hyperparameter.

*(Numerical Validation)* We validate this grid-particle design art from scientific computing by testing our network with different grid resolutions. As shown in Table 5.1, we tested the grid resolution of $8^3$, $16^3$, and $32^3$ on two datasets with 1024 and 2048 particles separately. We observed that a $16^3$ grid fits the 1024 dataset best and a $32^3$ grid fits the 2048 dataset best. By calculating the average $ppc$ for each case, we made a preliminary conclusion that the optimal $ppc$ is around 1.5-1.8. This also implies an optimal grid resolution for a point-set with $N$ particles to be $(ppc * N)^{1/3}$.

**PIC/FLIP**   *(Physical Intuition)* Temporal smoothness is key for developing a dynamic system to achieve its equilibrium state. PIC/FLIP obtains such smoothness by averaging weighted velocities between two adjacent time steps. *(Numerical Validation)* To highlight the role of this averaging, we compared the accuracy between PIC/FLIP and PIC only (no temporal averaging) on ModelNet10. We can see from Figure 7 that the model with PIC/FLIP quickly stabilizes to a high accuracy, outperforming the model with PIC only.

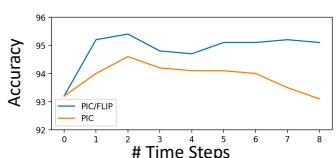

Figure 7: PIC/FLIP VS PIC

## 5.2 APPLICATIONS

**Classification** We tested our network on ModelNet40 (Z. Wu, 2015) and ModelNet10 for classification. We use a grid resolution $16^3$ to train both the networks. As shown in Table 3, our result outperforms the state-of-art on ModelNet10, noticeably surpassing those using grids. On ModelNet40, our result rivals DGCNN (92.8% v.s. 92.9%). But our parameter number is significantly smaller than DGCNN ($1M$ v.s. $21M$.)

**Segmentation** We tested our algorithm for object part segmentation on ShapeNet (Yi et al., 2016). We used a grid resolution and $32^3$ for training and testing. We

Table 3: Classification on ModelNet.

| Method | Input | ModelNet10 | ModelNet40 |
|---|---|---|---|
| SO-Net | 2048 pnts | 94.1 | 90.9 |
| PCNN | 1024 pnts | 94.9 | 92.3 |
| PointNet | 1024 pnts | - | 89.2 |
| PointGrid | 1024 pnts | - | 92.0 |
| DGCNN | 1024 pnts | - | **92.9** |
| PointCNN | 1024 pnts | - | 92.5 |
| PointNet++ | pnts, nors | - | 91.9 |
| SpiderCNN | pnts, nors | - | 92.4 |
| O-CNN | octree, nors | 91.0 | 86.5 |
| VoxNet | grid ($32^3$) | 92.0 | 83.0 |
| Kd-Net | kd-tree | 94.0 | 91.8 |
| FPNN | grid | - | 87.5 |
| MRCNN | multi-level vox | 91.3 | 86.2 |
| **Ours** ($16^3$) | 1024 pnts | **95.4** | 92.8 |

showed the state-of-art performance of our approach in Table 4. Since the category of each input object is known beforehand, we trained separate models for each category. Note that we only compared with point-based methods that had similar input (points or/and normals) as ours. It can be seen that we outperform all the state-of-art with less parameters ($1.1M$, 2 time steps) Some examples animating the segmentation process can be seen in Figure 6.

Table 4: Segmentation results on ShapeNet.

| Method | input | mIoU | aero | bag | cap | car | chair | ear phone | guitar | knife | lamp | laptop | motor | mug | pistol | rocket | skate board | table |
|---|---|---|---|---|---|---|---|---|---|---|---|---|---|---|---|---|---|---|
| PointNet | 2k pnts | 83.7 | 83.4 | 78.7 | 82.5 | 74.9 | 89.6 | 73.0 | 91.5 | 85.9 | 80.8 | 95.3 | 65.2 | 93.0 | 81.2 | 57.9 | 72.8 | 80.6 |
| PCNN | 2k pnts | 85.1 | 82.4 | 80.1 | 85.5 | 79.5 | 90.8 | 73.2 | 91.3 | 86.0 | 85.0 | 95.7 | 73.2 | 94.8 | 83.3 | 51.0 | 75.0 | 81.8 |
| Kd-Net | 4k pnts | 82.3 | 80.1 | 74.6 | 74.3 | 70.3 | 88.6 | 73.5 | 90.2 | 87.2 | 81.0 | 94.9 | 57.4 | 86.7 | 78.1 | 51.8 | 69.9 | 80.3 |
| DGCNN | 2k pnts | 85.1 | 84.2 | 83.7 | 84.4 | 77.1 | 90.9 | 78.5 | 91.5 | 87.3 | 82.9 | 96.0 | 67.0 | 93.3 | 82.6 | 59.7 | 75.5 | 82.0 |
| PointCNN | 2k pnts | 86.1 | 84.1 | **86.4** | 86.0 | 80.8 | 90.6 | **79.7** | 92.3 | **88.4** | **85.3** | 96.1 | 77.2 | 95.2 | 84.2 | **64.2** | **80.0** | 82.9 |
| PointNet++ | pnts, nors | 85.1 | 82.4 | 79.0 | 87.7 | 77.3 | 90.8 | 71.8 | 91.0 | 85.9 | 83.7 | 95.3 | 71.6 | 94.1 | 81.3 | 58.7 | 76.4 | 82.6 |
| SO-Net | pnts, nors | 84.9 | 82.8 | 77.8 | 88.0 | 77.3 | 90.6 | 73.5 | 90.7 | 83.9 | 82.8 | 94.8 | 69.1 | 94.2 | 80.9 | 53.1 | 72.9 | 83.0 |
| SpiderCNN | pnts, nors | 85.3 | 83.5 | 81.0 | 87.2 | 77.5 | 90.7 | 76.8 | 91.1 | 87.3 | 83.3 | 95.8 | 70.2 | 93.5 | 82.7 | 59.7 | 75.8 | 82.8 |
| SPLATNet | pnts, img | 85.4 | 83.2 | 84.3 | **89.1** | 80.3 | 90.7 | 75.5 | 92.1 | 87.1 | 83.9 | 96.3 | 75.6 | 95.8 | 83.8 | 64.0 | 75.5 | 81.8 |
| **Ours** ($32^3$) | 2k pnts | **86.2** | **84.4** | 83.8 | 85.7 | **81.7** | **91.1** | 74.7 | 91.7 | 87.2 | 84.9 | **96.4** | 72.2 | **95.9** | **84.3** | 58.5 | 75.1 | **83.4** |

## 6 DISCUSSION AND CONCLUSION

This paper presents a new perspective in treating the point cloud learning problem as a dynamic advection problem using a learned background velocity field. The key technical contribution of the proposed approach is to jointly define the point cloud learning problem as a flow advection problem in a world space using a static background grid and the local space using moving particles. Compared with the previous hybrid grid-point learning methods, e.g. two-way coupled particle-grid schemes (Su et al., 2018; Atzmon et al., 2018; Liu et al., 2019), our approach solves the learning problem from a dynamic system perspective which accumulates features in a flow field learned temporally. The coupled Eulerian-Lagrangian data structure in conjunction with its accommodated interpolation schemes provide an effective solution to tackle the challenges regarding both stencil construction and feature evolution by leveraging a numerical infrastructure that is matured in the scientific computing community. On another hand, our approach can be thought of as an exploration in creating a new physical reservoir motivated by continuum mechanics in order to find alternative solutions for the conventional point cloud processing networks. Thanks to the low-dimensional physical space and the large time step our network allows, our learning accuracy rivals the state-of-the-art deep networks such as PointCNN (Li et al., 2018b) and DGCNN (Wang et al., 2019) while using significantly fewer network parameters ($4\%$ to $25\%$ in our comparisons). Our future plan is to scale the algorithm to larger data sets and handle more complex point clouds with sparse and adaptive grid structures.

## 7 ACKNOWLEDGEMENT

This project is support in part by Dartmouth Neukom Institute CompX Faculty Grant, Burke Research Initiation Award, and NSF MRI 1919647. Helen Lu Cao is supported by the Dartmouth Women in Science Project (WISP) and Undergraduate Advising and Research Program (UGAR).

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

# A   PERFORMANCE ON S3DIS

In this part, we further discuss our algorithm and its performance on the large-scale S3DIS dataset (Armeni et al., 2017). Unlike ModelNet and ShapeNet, the S3DIS consists of colored point clouds collected from real world. We train on the area 1,2,3,4,6 and test on the area 5. We make some modifications on our network structure to better fir this dataset.

1. We set the number of time steps to 4 instead of 2.
2. To allow large number of time steps (deeper networks), we replace our original MLP with the ResNet blocks (He et al., 2015).
3. We use two more MLPs to encode the initial features on each point.
4. We scale the point cloud into the space $[-0.9, 0.9]^3$ because the data contains many points on the border plane, such as ceiling and floor.

The performance on the S3DIS is in the table 5.

Table 5: Segmentation results on S3DIS.

| Method | mIoU | ceiling | floor | wall | beam | column | window | door | table | chair | sofa | bookcase | board | clutter |
|---|---|---|---|---|---|---|---|---|---|---|---|---|---|---|
| PointNet | 41.09 | 88.80 | 97.33 | 69.80 | 0.05 | 3.92 | 46.26 | 10.76 | 58.93 | 52.61 | 5.85 | 40.28 | 26.38 | 33.22 |
| SPGraph | 58.04 | 89.35 | 96.87 | 78.12 | 0.00 | **42.81** | 48.93 | 61.58 | **84.66** | 75.41 | **69.84** | 52.60 | 2.10 | 52.22 |
| SegCloud | 48.92 | 90.06 | 96.05 | 69.86 | 0.00 | 18.37 | 38.35 | 23.12 | 70.40 | 75.89 | 40.88 | 58.42 | 12.96 | 41.60 |
| PCCN | **58.27** | 92.26 | 96.20 | 75.89 | 0.27 | 5.98 | **69.49** | **63.45** | 66.87 | 65.63 | 47.28 | **68.91** | 59.10 | 46.22 |
| PointCNN | 57.26 | 92.31 | 98.24 | **79.41** | 0.00 | 17.60 | 22.77 | 62.09 | 74.39 | **80.59** | 31.67 | 66.67 | **62.05** | **56.74** |
| **Ours** | 54.37 | **92.41** | **98.28** | 74.88 | **0.68** | 18.74 | 45.36 | 48.70 | 74.11 | 78.50 | 35.60 | 57.02 | 38.84 | 43.73 |

From the table we can see that the result obtained by AdvectiveNet is comparable to the states of the art (with the highest mIoU in ceiling, floor, and beam), though it is less impressive to the performance on ModelNet and ShapeNet. We observe that, in the S3DIS, the relative positions of different parts are more flexible and less structured compared to ModelNet and ShapeNet. For example, in ShapeNet, the wings of the airplanes are always on the two sides of the fuselages. Hence, we would interpret our performance on ShapeNet thanks to the ability in detecting intrinsic structures underlying the relative positions of the parts. The tendency to focus on relative positions also explains why our algorithm outperforms the states of the art on detecting ceiling and floor (they are always on the two sides of the rooms).

# B   IMPLEMENTATION DETAILS

We follow the data augmentation methods in (Li et al., 2018b). We use dropout ratio 0.3 on the last fully connected layer before class score prediction. The decay rate for batch normalization starts with 0.5 and is gradually decreased to 0.01. We use adamw optimizer (Loshchilov & Hutter, 2017) with initial learning rate 0.001, weight decay rate 0.005, momentum 0.9 and batch size 32. The learning rate is multiplied by 0.8 every 20 epochs. We train the model for 200 epochs. We use the label smoothing techique (Pereyra et al., 2017) with confidence 0.8. We use the grid size 16 and 32 for classification and segmentation, respectively.

