# OpenReview forum: "AdvectiveNet: An Eulerian-Lagrangian Fluidic Reservoir for Point Cloud Processing     "
_ICLR.cc/2020/Conference — Accept (Poster)_

### Official Review · AnonReviewer2 · 2019-10-21
**Official Blind Review #2**

**Rating:** 6

**Review:**

The paper presents a method for point-based learning that is inspired by a hybrid Eulerian-Lagrangian fluid simulation method. The work first explains how the simulation algorithm is mapped to the learning problem: MLPs are employed to learn sets of particle based features which are mapped to a Eulerian grid. A second MLP infers a particle based velocity, which is likewise mapped to the grid and used to advect the grid quantities. This is repeated for a certain number of steps to obtain final positions. The "warped" features are then projected back onto the particles to solve, e.g., a classification task. In contrast to a typical flow solver, the motion can be divergent, i.e., not necessarily conserves volume.

The paper presents a brief ablation study for number of iterated steps, grid size and point count, before presentation two comparisons with existing baselines.

Overall, I found the idea to employ FLIP for Lagrangian learning tasks novel and very interesting. Unfortunately, the paper (as mentioned in the text) only contains only a somewhat preliminary study. The method does not yield clear gains over previous work, but rather a similar performance for classification and segmentation of ShapeNet and S3DIS data is shown. Given the fairly complicated construction, I think it would be important to actually show improvements at least for specific learning tasks. Several of the deformations shown in figure 5 and 6 are also not really intuitive

Also, on second sight, I don't fully understand the motivation for employing and learning a grid based deformation. The grids seem to inherently limit the spatial extent of the point clouds, and the features that can be resolved. Features smaller than a grid cell will essentially "stick together", and can't be separated. It's also not obvious how to choose parameters such as the number of time steps. Intuitively, I'd expect the method to "converge" to a position for a larger number of steps.

To conclude, the direction this paper takes is certaily new and interesting, but the preliminary results in combination with the complexity and limitations introduced by the grid-based representation make me hesitant to recommend accepting this paper in its current form. (The nine pages also contribute to this assessment.)


**Experience Assessment:**

I have published in this field for several years.

**Review Assessment: Checking Correctness Of Derivations And Theory:**

I assessed the sensibility of the derivations and theory.

**Review Assessment: Checking Correctness Of Experiments:**

I carefully checked the experiments.

**Review Assessment: Thoroughness In Paper Reading:**

I read the paper thoroughly.

---

### Official Review · AnonReviewer1 · 2019-10-22
**Official Blind Review #1**

**Rating:** 6

**Review:**

The paper is about using classical PIC/FLIP scheme in Computational Fluid Dynamics for solving the learning problem of 3D object detection and segmentation. In general, there are extrinsic CNNs like the Vox net etc. which look for global features which the authors refer to as Eulerian formulation of the data representation, and there are intrinsic CNNs like the GCN(graph convolutions), Point nets etc. which look for localized neighborhood information which the authors refer to as Lagrangian formulation. The authors acknowledge that hybridizing the extrinsic CNNs and intrinsic CNNs is not new and several works are cited. The key contribution is to look at this problem from the perspective of PIC/FLIP scheme which has been used in CFD for decades.

The idea is very nice, well describes and quite novel in my opinion. I really liked the adoption of classical CFD approaches in learning. This provides a very interesting perspective to 3D deep learning.

However, the papers struggles to demonstrate why the 3D deep learning community would adopt this approach. The results are not that conclusive. The algorithm works (understood well from the ablation study). However, the performance of the proposed approach is at best comparable to some of the state-of-the-art methods such as PointCNN or SE-Net. The authors need to clarify what potential advantages could there be other than accuracy (if any) such that the community uses the proposed method.

Also, the grids used in the study are too low to make any conclusive remarks on what happens when dealing with higher resolutions of grid. The authors themselves acknowledge the limitation of not being able to go higher in resolution of grid. Interestingly, such limitations of CFD has recently motivated the community to explore deep learning based fast and agile surrogates for computationally tractable approaches.

Some of the new works in 3D object recognition and segmentation such as  Deep SDF(https://arxiv.org/abs/1901.05103), AtlasNet(https://arxiv.org/abs/1802.05384),  Deep Level Sets (https://arxiv.org/abs/1901.06802), occupancy networks (https://arxiv.org/pdf/1812.03828v1.pdf), http://openaccess.thecvf.com/content_cvpr_2018/html/Yu_PU-Net_Point_Cloud_CVPR_2018_paper.html, Adaptive O-CNN (https://dl.acm.org/citation.cfm?id=3275050), https://arxiv.org/abs/1805.12254, 3D Point Capsule Networks, http://t.cvlibs.net/publications/Niemeyer2019ICCV.pdf etc. can be compared with or at least contrasted in the related works.

In summary, I really liked the algorithmic idea, but skeptical about its practical relevance from the results.

**Experience Assessment:**

I have published in this field for several years.

**Review Assessment: Checking Correctness Of Derivations And Theory:**

I assessed the sensibility of the derivations and theory.

**Review Assessment: Checking Correctness Of Experiments:**

I assessed the sensibility of the experiments.

**Review Assessment: Thoroughness In Paper Reading:**

I read the paper at least twice and used my best judgement in assessing the paper.

---

### Official Review · AnonReviewer3 · 2019-10-28
**Official Blind Review #3**

**Rating:** 6

**Review:**

The paper addresses the task of learning with point clouds for semantic labeling (classification and segmentation). The authors propose a novel point-based architecture based on viewing the learning process as an advection in the 3D space. This formulation aims at an explicit connection the two formulations for learning with point clouds, the first being focused on points (Lagrangian formulation), the second on the regular spatial grid not necessarily coinciding with points (Eulerian formulation). While the connection between the two formulation is known in the literature, the paper does a good overview of the relevant work and highlights the interplay between the two treatments for learning, which is valuable to the reader. The proposed view of learning with point clouds is, as far as I know, novel.

With the proposed learnable operations, the authors are able to efficiently learn the functions defined in 3D space, such as the semantic class labels. The operations include transferring the features between the grid and the point cloud, advection, and interpolation, all implemented in a unified learnable model.

The architecture is evaluated on classification and segmentation tasks with common datasets, where it performs on par with existing methods. While the experimental evaluation does not indicate that the proposed method is a new state-of-the-art, it convincingly validates that the proposed method is capable of learning powerful enough representations.

I believe the paper should be accepted for publication, as (1) the proposed method is generally novel while it bases on solid and well-known foundations, (2) the experimental validation is sufficient to demonstrate the capabilities of the approach.

**Experience Assessment:**

I have read many papers in this area.

**Review Assessment: Checking Correctness Of Derivations And Theory:**

I assessed the sensibility of the derivations and theory.

**Review Assessment: Checking Correctness Of Experiments:**

I assessed the sensibility of the experiments.

**Review Assessment: Thoroughness In Paper Reading:**

I read the paper at least twice and used my best judgement in assessing the paper.

---

### Author Response · Authors · 2019-11-14
**Rebuttal and Revised Submission**

Dear Reviewers,

Thank you for all the constructive comments and the valuable feedback. We are very pleased to address your concerns and incorporate your suggestions by making several main updates (see the bullets below) in the revised manuscript. All the changes are colored as blue in the new version.

# MAIN CHANGES
- A local grid-particle interpolation scheme;
- Scalable grid resolution (with new 16^3 and 32^3 results);
- State-of-the-art accuracy (compared with PointCNN and DGCNN);
- Simplified network architecture;
- More convergence and ablation tests;
- Suggested reference;
- Physical intuitions behind design decisions;
- Condensed pages (to 8).

SCALABLE GRID:
The problem of scalability was addressed by our new implementation of a local trilinear interpolation stencil. Thanks to this software engineering effort, we are able to test the performance of grids with different resolutions and identify the best fit for a point-set. We discovered that the performance of the particle-grid structure is correlated to the average number of particles in cell (ppc), which aligns with the empirical experience of using PIC/FLIP engineering. Our experiments indicate that the best learning performance lives with ppc=1.5, implying the best grid size to be 16^3 for a 1024 point-cloud and 32^3 for a 2048 one (see Section 5.1 for details).

ROLE OF GRID IN A HYBRID REPRESENTATION
The primary role of a background grid is to enable an efficient construction of the various differential operators (in our case convolution) on-the-fly without fitting a local parameter space from the potentially noisy Lagrangian samples. The existence of such grid-based differential stencils enables fast numerical implementation, efficient data access patterns thanks to its cache-friendly data storage. From the learning point of view, a grid can be regarded as a perceptron for the relation on a coarse-level (similar to the role of the furthest point step in PointNet++ or the KNN step in DGCNN), but with a more regularized and efficient spatial representation. This representation complements the local particle-based operators.

NETWORK ARCHITECTURE
We simplified the network architecture by merging the interpolation and advection modules to further approach the essence of its physical model (see Section 4 for details). This simplification leads to a more economical computing model w.r.t both memory and time.

STATE-OF-THE-ART PERFORMANCE
The better network architecture, in conjunction with the improved interpolation stencil (see Section 4) and initialization conditions (see Section 3), leads to a better performance compared to our previous manuscript. Our current model with the improved implementations can rival the state-of-the-art approaches, in particular PointCNN and DGCNN, regarding both accuracy and memory efficiency. As shown in Table 4, the AdvectiveNet can beat PointCNN and DGCNN in approximately half of the test cases we have run, yet consuming only **4% - 25%** model parameters of the state-of-the-art. Currently, we evaluated the new version on most of the datasets and obtained very promising results (S3DIS is expected to finish in a few days). Hence, we believe this approach can become an effective tool for point-cloud processing with the state-of-the-art performance.


TEMPORAL EVOLUTION:
We conducted a series of further evaluations on the temporal discretization and observed that our method is stable with large time steps (see the Ablation Tests for details). We demonstrated such stability regarding both the temporal convergence (accuracy) and the spatial convergence (the final shape advected by different timesteps). We also gave more physical and numerical explanations for each of the tests.

OTHER MINOR COMMENTS:
Reviewer #1:
-- Potential advantages:
Besides its comparable accuracy performance, the low GPU memory consumption (1.1G GPU memory with batch size 16 and points 1024 on ModelNet10) and the low-dimensional feature space (projected on the physical space) enabling dynamic neighbor relations are the two main advantages of our Eulerian-Lagrangian approach.

-- Suggested reference:
We have incorporated the suggested references in our related work. Thank you!

Reviewer #2:
-- Role of advection flow
We use an Eulerian flow field to dynamically rebuild the neighbor relations in a physical space, in which case the incompressibility was not our major concern. But this is an interesting property to explore and we added it to our future work.

-- Intuitive deformation
We provided further comparisons in Figure 6 to demonstrate the shape convergence.

-- Spatial extent of grid
The boundary conditions will guarantee the coverage of the grid on the particles advected by the learned velocity field.

-- Particle sticking together
Particles won't stick together during advection because of the compatible resolutions between the Eulerian and Lagrangian degrees of freedom (see our discussion on the ppc number in Section 5.1).

---

### Public Comment · ~Duc_Anh_Nguyen1 · 2020-02-14
**Where is the result on S3DIS?**

Hi. Congratulations on the acceptance. The paper seems interesting. However, I couldn't find the result of segmentation on the S3DIS dataset. Could you please show me?
Also, have you tried your method on some other realistic dataset (for e.g., Scannet)?
Thanks.

---

> ### Author Response · Authors · 2020-02-24
> **Supplementary**
>
> Hi Duc,
>
> Thanks for your comments. We put the S3DIS results along with our brief analysis in a supplementary. Please see the latest updates. We did not test other realistic datasets such as Scannet. We found that some future investigation is needed to enable the network to handle the 'unstructured' point clouds (which potentially not follow the nature advection process). But we could conduct these tests if necessary. Thank you again for all your time and suggestions!
>
> Best,
> The authors

---

### Decision · Program_Chairs · 2019-12-19

**Decision:**

Accept (Poster)

**Comment:**

This paper treats the task of point cloud learning as a dynamic advection problem in conjunction with a learned background velocity field.  The resulting system, which bridges geometric machine learning and physical simulation, achieves promising performance on various classification and segmentation problems.  Although the initial scores were mixed, all reviewers converged to acceptance after the rebuttal period.  For example, a better network architecture, along with an improved interpolation stencil and initialization, lead to better performance (now rivaling the state-of-the-art) as compared to the original submission.  This helps to mitigate an initial reviewer concern in terms of competitiveness with existing methods like PointCNN or SE-Net.  Likewise, interesting new experiments such as PIC vs. FLIP were included.